# Zebrafish as an Orthotopic Tumor Model for Retinoblastoma Mimicking Routes of Human Metastasis

**DOI:** 10.3390/cancers14235814

**Published:** 2022-11-25

**Authors:** Nenad Maricic, Melanie Schwermer, Alexander Schramm, Gabriela Morosan-Puopolo, Petra Ketteler, Beate Brand-Saberi

**Affiliations:** 1Department of Anatomy and Molecular Embryology, Ruhr-University of Bochum, D-44801 Bochum, Germany; 2Institute of Anatomy and Molecular Neurobiology, Westfälische-Wilhelms University, D-48149 Münster, Germany; 3Department of Pediatrics III, University Hospital Essen, University Duisburg-Essen, D-45147 Essen, Germany; 4Department of Medical Oncology, West German Cancer Center, University Hospital Essen, D-45147 Essen, Germany

**Keywords:** retinoblastoma, malignant retinal tumor, childhood eye cancer, RB355, WERI-RB-1, Y79, orthotopic transplantation, zebrafish

## Abstract

**Simple Summary:**

Retinoblastoma is a rare malignant eye tumor with early childhood occurrence and high invasive potential. Animal models of retinoblastoma have multiple limitations, but recently published zebrafish models are promising. The transparent nature of zebrafish allows visualization of migrating cells in the living young fish under the microscope. Previous zebrafish models have analyzed only side views of the heads whereas the aim of our study is to analyze the timing and the metastatic trajectories of different retinoblastoma cell lines from two angles, the side view and dorsal view. In our zebrafish model, migrating retinoblastoma cells are found along the optic nerve and adjacent regions of the brain or its ventricles. These trajectories of migrating cells resemble the pattern of metastasis in human patients. The zebrafish model may facilitate pharmacological studies for treatment of retinoblastoma in the future. Our results provide new insights into the metastatic behavior of this complex tumor.

**Abstract:**

Background: Retinoblastoma (RB) is the most common eye cancer in children that has a high mortality rate when left untreated. Mouse models for retinoblastoma have been established but are time- and cost-intensive. The aim of this work was to evaluate an orthotopic transplantation model of retinoblastoma in zebrafish that also allows for tracking migratory routes and to explore advantages and disadvantages with respect to drug testing. Methods: Three fluorescence-labeled retinoblastoma cell lines (RB355, WERI-RB-1, Y79) were injected into the left eye of two-day-old zebrafish, while the un-injected right eye served as control. The migratory trajectories of injected retinoblastoma cells were observed until 8 days post injection (dpi), both in lateral and dorsal view, and measuring fluorescence intensity of injected cells was done for RB355 cells. Results: Time until the onset of migration and routes for all three retinoblastoma cell lines were comparable and resulted in migration into the brain and ventricles of the forebrain, midbrain and hindbrain. Involvement of the optic nerve was observed in 10% of injections with the RB355 cell line, 15% with Y79 cells and 5% with WERI-RB-1 cells. Fluorescence intensity of injected RB355 cells showed an initial increase until five dpi, but then decreased with high variability until the end of observation. Conclusion: The zebrafish eye is well suited for the analysis of migratory routes in retinoblastoma and closely mirrors patterns of retinoblastoma metastases in humans.

## 1. Introduction

Retinoblastoma is the most common eye cancer in children and arises from retinal progenitor cells during the first years of childhood. The incidence of retinoblastoma is at 1 in every 15,000 to 20,000 live births [1], which is equivalent to about 9000 newly diagnosed patients worldwide every year. Geographic regions with the highest prevalence, mostly developing countries, also have the highest mortality rate [2]. In those countries, patients commonly present with proptosis or clinical signs of metastatic disease already at diagnosis [3]. Metastatic retinoblastoma has poor survival rates despite aggressive multimodal treatment [4,5].

Retinoblastoma can affect one eye (unilateral) or both eyes (bilateral), and in rare cases also the pineal or suprasellar regions (trilateral retinoblastoma). Non-heritable retinoblastoma is characterized by inactivation of both alleles of the retinoblastoma gene (*RB1*, on chromosome 13q14). In this form, both wild-type alleles of *RB1* gene are inactivated in the tumor tissue, leading to a single unifocal tumor, whereas children with heritable retinoblastoma have a constitutional pathogenic variant in *RB1* and thus only require a mutation of the second allele to initiate cancer. Most children with heritable retinoblastoma develop bilateral disease. The *RB1* gene encodes the protein pRB, a key negative regulatory transcription factor playing a role in cell cycle transition from G1 to S phase by blocking the interactions of DNA-binding of E2 promotor binding factor (E2F) and Dimerization Partner (DP) transcription factors with DNA. Hyperphosphorylation of pRB in response to mitogen-stimulated cyclin-dependent kinase signaling releases it from E2F interactions, allowing E2F and DP to activate transcription of genes involved in G1/S progression [6]. Apart from *RB1* gene mutations, only a few recurrent genetic alterations have been linked to retinoblastoma development. In a small fraction (1.4% of unilateral retinoblastoma) no *RB1* mutation was found, but instead an amplification of the *MYCN* oncogene could be detected and these tumors showed an aggressive clinical course [7]. Although being rare, this tumor subtype could serve as a model to better understand the biology and clinical characteristics of aggressive retinoblastoma.

During local spread, retinoblastoma may produce intraocular seeds that detach and adhere below the retina (subretinal seeds) or float into the vitreous body (vitreous seeds). The tumor can spread beyond the eye by surpassing retinal boundaries and invading the choroid, finally reaching the blood supply. Another common route of metastasis is via the optic nerve towards the brain and the cerebrospinal fluid in the brain ventricle [8]. Modelling retinoblastoma metastasis in vivo is complicated as RB1 inactivation alone is not sufficient to induce retinoblastoma in mice [9]. Instead, combined loss of RB, p107 and p53 is required to produce mutant retinal progenitor cells that give rise to intraocular retinoblastoma with invasion of the tumor into the anterior eye chamber [10]. This is considered as a genuine mouse model of retinoblastoma.

In 2013, Jo et al. demonstrated orthotopic transplantation of retinoblastoma cells into the vitreous cavity of the zebrafish for screening of anticancer drugs. They showed a stable tumor growth for 4 days after injection. The authors demonstrated that treatment with carboplatin or melphalan reduced cell viability by 74% and 61%, respectively, after 4 days of treatment [11].

Another orthotopic zebrafish tumor model was reported using human and murine retinoblastoma cells that allowed for tracking of tumor cells for a time of ten days [12].

However, it remained to be evaluated if zebrafish models of retinoblastoma allow for faithfully recapitulating main trajectories of migration and the timing of tumor outgrowth and potential redistribution of cells that could be interpreted as mimicking metastases. To address these questions, we adapted and optimized the protocol suggested by Jo et al. [11] to prolong the interval for follow-up after tumor cell injection using three retinoblastoma cell lines RB355 (adherent cells), WERI-RB-1 (non-adherent cells) and Y79 (non-adherent cells). We demonstrate that tumor cells can be tracked in vivo up to nine days after injection. We here present evidence that orthotopic transplantation of human retinoblastoma allows to recapitulate metastatic routes using the zebrafish as a model.

## 2. Materials and Methods

### 2.1. Cell Lines and Cultivation

Established and well-characterized human retinoblastoma cell lines RB355 (adherent growth), Y79 (growth in suspension) and WERI-RB-1 (growth in suspension) [13,14] were used for all in vivo experiments. RB355 and Y79 are MYCN-amplified, while WERI-RB-1 has a gain of the MYCN oncogene (1.5–4-fold). RB355 cells were kindly provided by Professor Brenda Gallie (Department of Opthalmology and Vision Sciences, Children Hospital, Toronto, ON, Canada) [15,16]. Cell line Y79 (DSMZ-ACC 246) and WERI-RB-1 was purchased from the Leibniz-Institut DSMZ (German Collection of Microorganisms and Cell Cultures, Braunschweig). Cells were cultured in Dulbecco’s Modified Eagle Medium (DMEM 1x + GlutaMax-I, Invitrogen (ThermoFisher Scientific, Schwerte, Germany)) supplemented with 10% fetal calf serum (FCS, Invitrogen) in 100 mL cell culture flasks in a humidified atmosphere of 5% CO_2_ at 37 °C. The DMEM cell culture medium was further supplemented with 100 U/mL penicillin/streptomycin (Invitrogen), 2.5 mg/L amphotericin B (PAA Laboratories GmbH, Pasching, Austria) and 80 µL G418 (Carl Roth GMbH, Karlsruhe, Germany) per 10 mL medium. RB355 cells were detached from cell culture flasks with 1 mL Trypsin-EDTA solution/10 mL cell culture medium (Trypsin-EDTA Solution 0,25%, Gibco ThermoFisher Scientific, Darmstadt, Germany). Cell suspensions (10 mL) were centrifuged for 5 min and 600 g to obtain cell numbers compatible with injections into the zebrafish eye (approximately 10^6^ cells per mL). The identity of the cell lines was confirmed by Short tandem repeat STR analysis (German Collection of Microorganism and Cell Culture Braunschweig, Germany (Deutsche Sammlung von Mikroorganismen und Zellkulturen GmbH, DSMZ).

Concerning the results of Madreperla et al. (1991) [17], we could not reproduce the results of Madreperla et al. that RB355 is a subclone of Y79. Our results on the sequencing of two alleles that were often mutated in retinoblastoma and on the MYCN-status for 10 retinoblastoma cell lines are published in Schwermer et al. 2019, [13].

### 2.2. Generation of GFP Labeled Cells

The Gateway^®^pT-Rex^TM^-DEST30-GFP vector was transferred into retinoblastoma cells by electroporation. For cell transfection we used the Neon Transfection System (Invitrogen). Prior to electroporation, cells were centrifuged for 2 min and 125 G. Cells were counted and resuspended in OptiMEM medium (Invitrogen) to a density (cell concentration) of 0.5 × 10^6^ cells/10 µL. After adding 1 µg of plasmid (Gateway^®^pT-Rex^TM^-DEST30-GFP) per 10 µL medium, cells were electroporated by using three impulses with a duration of 20 ms and 1400 V (Microporator, Peqlab Biotechnologie, Erlangen, Germany). Transfected cells were initially cultured on petri dish with Dulbecco’s Modified Eagle Medium (DMEM 1x + GlutaMax-I, Invitrogen) supplemented with 10% fetal calf serum (FCS, Invitrogen) in a humidified atmosphere of 5% CO_2_ at 37 °C. One week after electroporation, GFP-positive cells were selected in medium containing Geneticindisulfate (G418, Carl Roth, 50 µg/mL). GFP-positive cells were microscopically verified and cultured cells were FACS-sorted to select the GFP-positive cell fraction for in vivo experiments.

### 2.3. Zebrafish Scheduling

Zebrafish (*Danio rerio* wildtype) were kept in 3-liter plastic boxes with 20 fishes (male and female) per box in an aquarium system (stand-alone unit V30 with PP module from Aqua Schwarz, Germany). The fishes were fed with dry food once per day and with *Artemia salina* live food twice per week. The pH value of the aquarium water was 7.5–8.0, and the dark/light cycle was set to 10/14 h. To get fish spawn, fish were transferred to spawn-boxes with a fine mesh, so that the eggs could sink through the mesh to the bottom of the spawn-boxes from where they were collected. Eggs were transferred in E3-medium and incubated in a warming cupboard at 28 °C. Twenty-eight hours after egg deposition and fertilization (28 hpf) the eggs were transferred to E3-medium with 0.2 mM Phenylthiourea (PTU) to bleach the eyes and developing melanocytes in the head and tail (E3-PTU medium). This medium was also used during the entire period after injection of retinoblastoma cells. E3-medium was prepared as 60× stock solution with 17.2 g NaCl, 0.76 g KCl, 2.9 g CaCl_2_ (2 H_2_O), 4.9 g MgSO_4_ (7 H_2_O) and 3 mL 0.01% methylene blue-solution in 1 L Aqua dest. PTU (from Sigma Aldrich Chemie GmbH, Taufkirchen, Germany) was prepared as 50× stock solution with 0.152 g/100 mL Aqua dest. (0.01 M) and then diluted 1:50 in E3-medium for incubation of zebrafish stages.

### 2.4. Cell Injection into the Eye

Zebrafish eggs were dechorionated 48 h post egg fertilization (48 hpf) with two forceps under a binocular microscope. Before cell injection, the dechorionated zebrafish were anesthetized with tricaine (0.042 mg/mL E3-medium) and then placed on a small sheet of wet paper in a Petri dish. Injection needles were prepared from glass capillaries (80 mm length, 1.55 mm diameter, from Hirschmann Laborgeräte GmbH, Eberstadt, Germany) with a needle puller to get needles with a tip of 0.02 mm in diameter. Needles were placed in a needle holder (Aspirator tube assemblies for calibrated microcapillary pipets from Sigma-Aldrich, A5177-5EA) with a rubber sleeve and a mouthpiece. Freshly prepared cell suspension (0.1–0.2 µL) was slowly injected into the left eye of the zebrafish using 70 to 90 cells (RB355 and WERI-RB-1 cells) or approximately 80 to 100 cells (Y79 cells). Successful injection was controlled by monitoring the green fluorescence of the cells. Cell injection into the left zebrafish eye was performed on twenty zebrafish for each of the cell lines. The un-injected right eye served as internal control and five completely un-injected zebrafish were used as negative control for each injection series. The experiments were approved by the German Ministry of Environment, Agriculture, Nature and Consumer Protection of North Rhine-Westphalia (LANUV) with the number 84-02.04.2016.A346.

### 2.5. Documentation of the Cell Injections

After cell injection, each zebrafish was separately transferred into an individual well of a 24-well plate together with 2.5 mL E3-medium with PTU. The injected eye of the fish was photographed directly after cell injection from the left head side (0 dpi) and on days one, two, five, seven and eight post injection (dpi) from left side and from dorsal with a magnification of 10× using a Leica BX165 binocular fluorescence microscope (the dorsal view was not photographed on the injection day; because of the yolk, the fishes could not be placed correctly) and in some cases with a Zeiss LSM 800 confocal microscope to get a better resolution. Medium containing PTU was also changed on these days. After dpi five, zebrafish were fed with very small portions of dry food in the wells.

### 2.6. Analysis of Fluorescence Changes and Localization of Tumor Migrations

As an indicator for the tumor increase or decrease, the documented photographs on different days post injection were analyzed using Image J (W. Rasband, NIH) by marking the tumor cell margin with a line and calculating the intensity for GFP-fluorescence (Analyze/Color Histogramm/value for green) and the area of the marked tumor (Analyze/Measure/value for area). Statistical analysis of standard deviation for GFP-fluorescence intensity was accomplished using MS Excel (Microsoft). Fluorescence quantification was made from the 2D images, because to do this with z-stack at the confocal microscope with each zebrafish on a daily basis would harm the young fish as it is time-consuming and involves placing cover slips over the zebrafish.

To identify the regions which were invaded by the tumor cells, we used the Max Planck Zebrafish Brain Atlas (mapzebrain) [18], the Atlas of Early Zebrafish Brain Development (Mueller and Wullimann, 2016) [19], where head sections are shown and the Cellular Resolution Atlas of the Larval Zebrafish Brain (Kunst et al., 2019) [20]. To confirm the trajectory of migrations into the brain ventricles, we injected a solution of methylene blue directly into the hindbrain ventricle (adapted from Lowery and Sive, 2005 [21]) and Gutzman and Sive, 2009 [22] and also in a second experiment we injected RB355 cells into the hindbrain ventricle and compared the results with the localization of metastasized cells after injection into the eye. For a further analysis of cell metastasis after injection into the eye we prepared hematoxilin-eosin (H.E.) stained paraffin-embedded microtome sections (10 micometer) at the day of cell injection and at the time of 2, 5, 7 dpi. Information about zebrafish eye development and zebrafish retina layers were obtained from Eastlake, 2017 [23], Gestri, 2012 [24], Bibliowicz, 2011 [25].

## 3. Results

### 3.1. Orthotopically Injected Retinoblastoma Cells Are Viable in the Zebrafish Eye and Migrate into the Brain Ventricles or along the Optic Nerve into the Brain

Retinoblastoma cell injection into the vitreous cavity of zebrafish with the cell line RB355 showed a tumor cell growth, detected by an increase in fluorescence intensity from 100% (1 day post injection, (dpi)) to 144% (5 dpi) for the lateral sideview and a tumor cell growth to 122% for the dorsal view (5 dpi). However, at 8 days post injection the tumor size decreased to 45% (lateral view) and 42% (dorsal view). An increase of the tumor size in the first few days but then a rapid decrease could also be seen for the WERI-RB-1 and Y79 cell lines. For 20 injections per cell line, the mean value of the observation times until no tumor was detectable anymore were 6.6 dpi and 5.9 dpi for the RB355 and Y79 cell lines, respectively. Mostly, the WERI-RB-1 cell line was only observable for less than five days (mean 4.2 days for 20 injections), because of fading fluorescence or cell death. The mean value for the first observable metastasis was 1.6 dpi for RB355 cells, 2.4 dpi for Y79 cells and 1.2 for WERI-RB-1 cells (Appendix A). The damage to the eye after a very careful injection technique was only subtle and could not be detected over the next 1–2 days. In all three cell lines from 20 injected zebrafish, only one zebrafish died during the first day post injection.

Systematic analysis of the metastasis localization at the maximum observable time of the tumor (Appendix A) was performed for the 20 zebrafish injected with the three different cell lines. For the analysis of cell migration in the fish, migrations into the ventricle, into brain tissue or to the margin of the eye (short migration) were numbered. If a fish has a migration into the ventricle and into brain tissue, cells of both migrations were counted, so the number of migrated cells can be more than 20 per group. If a fish has a main migration into ventricle or brain tissue and only few cells at the eye margin, only the main migration was counted.

After 5 to 8 dpi, RB355 cells were found in the brain ventricles (6 from 20 cases, 30%), in other brain regions (7 from 20 cases, 35%) or near the border of the injected eye (6 from 20 cases, 30%). Retinoblastoma cells remained viable in the eye without migrating in one of 20 injections (5%) of RB355 cells; one case showed migrations into both the ventricle and other brain regions (Appendix A). Y79 cells migrated in eight fish into the ventricles (40%), in five (25%) into other brain regions, and in seven fish (35%) only to the border of the injected eye and in one case (5%) no migration occurred at all. Two cases showed ventricle and brain metastases (Appendix A) WERI-RB-1 cells migrated in two fish into the ventricles (10%) and in six fish into other brain regions (30%), and nine fish (45%) showed migrations only to the border of the left eye (cells at the border of the left eye were only counted as short migrations, when this was the only metastasis) and in two cases no migration occurred (10%, Appendix A).

An important trajectory of retinoblastoma cells was along the optic nerve into the brain in the first five days after injection. The migration of retinoblastoma cells out of the eye along the optic nerve could be postulated from the presence of a continuous track of fluorescent cells (Figure 1), that can be observed in all three cell lines. The presence of such a continuous track of fluorescent cells and therefore a migration along the optic nerve can be observed in 10%, 15% and 5% of fish injected with RB355 cells, Y79 cells and WERI-RB-1 cells, respectively. However, in cases that show no continuous track of cells, but cells that invaded the brain, the trajectory along the optic nerve is likely because a propagation through the blood vessels could not be found. There were no differences observable regarding the number of eyes with cell migration among the three different cell lines, but fluorescence intensity was strongest in RB355 cells allowing for a more detailed observation beyond 5 dpi, whereas the WERI-RB-1 cell line was mostly not observable beyond five dpi. Thus, we focused on the results for fluorescence intensity on the RB355 cell line. In most eyes with RB355, nearly all cells migrated out of the eye until five days after cell injection and no or only a few remaining cells were visible in the injected left eye. Some tumor cells migrated out of the injected eye through the brain towards the contralateral eye, but did not enter the second eye during the observation period of 9 dpi. In none of the experiments, tumor cells migrated to regions other than the brain or ventricles and the remainder of the fish body was always free of tumor cells until the end of the observation period. From these results, we postulated that tumor cells did not metastasize via the blood vessels. Of note, the fluorescence of the tumor cells faded after five days and tumor cells were only visible in rare cases after day eight post injection (Figure 2).

### 3.2. Fading GFP Fluorescence Limited the Observation Time Period

RB355 cell line was the best observable cell line because of the highest fluorescence intensity and the longest observation time (Appendix A). The mean GFP-fluorescence intensity received by analysis of the photographs of 19 zebrafish embryos injected with RB355 cells increased from 0 dpi until 5 dpi and then decreased until 8 dpi in both views, the left side view and the dorsal view (Figure 2).

Analysis of 20 fish with injection of RB355 cells showed that metastasis of retinoblastoma cells was found already after one day (13 of 20 eyes), in other cases after two days (3 of 20 eyes), or 5 days (two cases) and in one case no metastasis was found until the end of observation. The observation time of these 20 injections was 9 dpi (one case), 8 dpi (eight cases), 7 dpi (one case) and 5 dpi (nine cases) until the fluorescence intensity faded.

### 3.3. Migration of Retinoblastoma Cells Was Detectable Earlier in Dorsal than in Lateral View

Visualization of the timing and the trajectory of metastasis showed a similar pattern of metastasis in all three cell lines. However, within all three cell lines, retinoblastoma cells migrated via different routes and to different locations in the brain (ventricles, forebrain, midbrain, hindbrain).

For all three cell lines, migrations outside the eye were first visible in the dorsal view. Migration of the WERI-RB-1 cells out of the eye could be seen at 1 dpi in the dorsal view (Figure 1F and Figure 3E) and after 2 and 5 dpi in the lateral view (Figure 1C,D and Figure 3C,D). After 5 dpi, only a few cells remained in the eye while the majority of cells migrated to the brain (Figure 1D,H and Figure 3D,G). Nearly all WERI-RB-1 cells had migrated out of the eye to the posterior forebrain and anterior hindbrain at 5 dpi, while only few cells remained at the margin of the injected eye (Figure 1H and Figure 3G). In Figure 1F,G the trajectory of the cells can be postulated along the optic nerve, because of the beaded formation of metastatic cells from the injected eye to the brain. Precisely, in the second case, the migration did not start at the optic nerve; it started at the anterio-cranial margin of the eyeball (Figure 3F). Visualization of tumor cell migration in lateral and dorsal view emphasized that consideration of only one of these views can lead to misinterpretations of the migration start time, localization and trajectory of migrating cells, because the migration along the optic nerve can be postulated only from the dorsal view. Using GFP-labeled retinoblastoma cells, the fluorescence of the tumor cells became weak after five days and on day eight tumor cells were only visible in selected zebrafish (Figure 3H (8 dpi) compared to Figure 1H (5 dpi)).

In a few eyes, the retinoblastoma cells migrated out of the injected eye through the brain close to the contralateral eye, but did not enter the second eye until day 9 post injection. Migration of tumor cells along the optic nerve towards and to the contralateral eye was supported by five scans of a z-stack through the zebrafish head in the dorsal view at 1 dpi of 80–100 Y79 cells injected in the left eye (Figure 4A–E).

### 3.4. All Three Retinoblastoma Cell Lines Metastasized to the Brain Ventricles

From 5 dpi until 9 dpi retinoblastoma cells of all three cell lines migrated further towards the anterior region of the head (forebrain) and into the posterior region of the head (hindbrain). This seems to be the endpoint of migrations when the cells migrated along the optic nerve. The second route of migrations observable in all three cell lines was the trajectory into the brain ventricles. The pattern of distribution of the invasive retinoblastoma cells appeared to mimic the shape of the brain ventricles of the forebrain, midbrain and hindbrain (Figure 5B–D,F–H). Dissemination of the retinoblastoma cells into the zebrafish brain ventricles was confirmed by the similarity of the pattern of tumor cell migration and the pattern of methylene blue distribution after injection of a solution of methylene blue (Figure 6A,B) directly into the hindbrain ventricle and injection of RB355 cells into the ventricle (Figure 6C,D) (adapted from Lowery and Sive, 2005 [21]) and Gutzman and Sive, 2009 [22]. The higher resolution of the confocal Zeiss LSM 800 microscope enabled us to visualize individual cells in the cluster of the tumor migrations (Figure 7) in the brain ventricles. Precisely, this appeared to be a second route of migrations, because cells were never observable in the brain ventricles, when they were observed along the optic nerve. Additionally, migrations in the brain ventricles were observable at a very early time point, in some cases at 2 dpi.

### 3.5. Migration of Retinoblastoma Cells Was Confirmed by Hematoxylin-Eosin Staining

Migration of RB355 cells along the optic nerve to the brain after injection in the vitreous cavity was confirmed by fixation of the zebrafish and hematoxylin-eosin (H.E.) standard section staining. In the higher magnification (40×) the different retina layers and the exit of the optic nerve could be distinguished (Figure 8B). At the day of cell injection, RB355 cells penetrated the outer layers of the retina (Figure 8C) and at 2 dpi, the cells were detected in the forebrain (Figure 8D). RB355 cells reached the forebrain ventricle at 5 dpi (Figure 8E) and tumor cells could be seen in the forebrain ventricle and near the region of the optic chiasma at 7 dpi (Figure 8F).

Further details of migrating RB355 cells can be seen in Figure 9 with higher magnification from slides of Figure 8C–F (or neighboring slides). Figure 9D showing clearly injected RB355 cells near the axons of the optic nerve (marked with the asterisk).

## 4. Discussion

The aim of our study was to establish a retinoblastoma tumor model in the zebrafish and to describe the migration behavior of three different retinoblastoma cell lines (RB355, WERI-RB-1, Y79) after injection into the vitreous cavity of the zebrafish eye. The migration of these different cell lines after injection of 80 to 100 retinoblastoma cells into the left zebrafish eye was monitored until the fading of the fluorescence (maximum 9 dpi). The adherent cell line RB355 did not show a marked difference with respect to the capacity of the cells to migrate in comparison to the suspension cell line Y79. All three cell lines started to migrate out of the injected eye after one to three dpi, and after five dpi, most of the injected cells could be found in the forebrain, midbrain or hindbrain. Our study is the first one that focuses in detail on the migration route of the retinoblastoma cells in the zebrafish along the optic nerve and into the ventricles. In some cases, but in all three analyzed cell lines, we observed the migration of the cells along the optic nerve (shown in Figure 1 and Figure 3 and in detail with the confocal Zeiss LSM 800 microscope in Figure 4), and we postulated this being the main trajectory of the cells because hematological spread was not observed. The trajectory of the cells along the optic nerve can only be shown when a beaded formation of cells can be seen and this may be only for a short time (Figure 1 and Figure 4). Because the optic nerve of a young zebrafish is very thin and does not plane in one layer to the optic chiasma, it is mostly not possible to get the whole optic nerve in one slide. Therefore, injected retinoblastoma cells can only be shown for a short distance near the optic nerve in the slides. For this reason, we used both methods, the H.E.-stained slides (Figure 9D) and confocal microscopy on GFP-transfected cells with the z-stack of five planes (Figure 4F), to demonstrate the injected cells along or near the optic nerve. A similar route of migration along the optic nerve and into the brain ventricles is common in children with retinoblastoma [8].

A migration of tumor cells out of the brain or brain ventricles into other regions of the zebrafish body or tail could not be detected in any of the three retinoblastoma cell lines and until the end of the analysis time of nine days. Furthermore, we could not demonstrate any hematological metastases in all analyzed cases. These results are in strong contrast to the results of Chen et al., 2015 [12], who postulated hematological metastasis using SJmRBL-8 cells and RB355 cells that were orthotopically injected into zebrafish. Hematogenous metastases of the tumor cells seem unlikely to be the way of metastasis in the first days after injection, because the diameter of the head vessels is probably too small for a tumor cell transport into the zebrafish trunk and tail. However, we could confirm results of Chen et al. [12] regarding stable mean fluorescence intensity of the primary tumor area from the time of injection (0 dpi) up to five days post injection (5 dpi). Furthermore, in our experiments there was a constant decrease in the fluorescence intensity after the fifth day post injection and also in the experiments of Chen et al. there was a constant decrease in the tumor area after the sixth day until day ten post injection. The reason for the decrease in the fluorescence intensity could be loss of tumor cells by cell death or because the cells segregated from each other and therefore were not as bright as in a cluster of cells, or the fluorescence of the cells faded out. In line with our findings, Chen et al., 2015 [12] described in their orthotopic fli1:EGFP transgenic zebrafish tumor model a decrease in fluorescence after four days post injection of the primary tumor area. Their analysis focused on migration of the cells within the developing vessel system that are depicted in green fluoresce in the fli1:EGFP transgenic zebrafish. We could not reproduce these results as we found no migration of cells in the tail of the fish. Possible reasons for this discrepancy could be the different zebrafish model used (zebrafish wildtype vs. transgenic zebrafish) and different time for cell implantation. In the fli1:EGFP transgenic zebrafish, the green fluorescence of the vessels may mask the optic nerve.

Jo et al. (2013) [11] also found a strong increase in fluorescence intensity from 0 dpi until 1 dpi and afterwards the fluorescence intensity only slightly increases from 1 dpi until 4 dpi, which was the total time period of their experiment. Our own findings corroborate these results, because we also find an increase in the fluorescence intensity until 5 dpi, but thereafter a decrease at 7 and 8 dpi. The fading of the fluorescence seems to be the most likely reason; cells are no longer observable after a longer time because in the H.E.-stained sections, injected cells could be observable also on day seven post injection (Figure 8F and Figure 9D).

Comparing the three retinoblastoma cell lines used in our study, we obtained the best results with the RB355 cell line, because it showed the highest fluorescence intensity and could be observed for the longest time, but the migration characteristics of all three cell-lines were comparable.

In 2019, Asnaghi et al. published two articles [26,27], in which they analyzed the function of ACVR1C/ALK7, a type I receptor of the TGF-ß family and its ligands Nodal, Activin A/B and GDF3. They found a three-fold increase of the receptor ACVR1C in invasive retinoblastomas, which invaded the optic nerve compared to non-invasive tumors [26]. Knockdown of the ligand Nodal with shRNA resulted in reduction in phosphorylated SMAD-2, which functions as downstream signaling of the ACVR1C pathway. Downregulation of Nodal also showed a reduction in the epithelial-to-mesenchymal transition (EMT) markers ZEB1 and snail and a reduction in migration of cells in the transwell invasion assay. Further, they showed a 34% reduction in the so- called minimum bonding sphere (MBS) at 4 dpi, which represented the migration of Y79 cells injected in the zebrafish eye [27]. Both articles from Asnaghi et al. analyzed the pathways that are involved in cell migration of Y79 and WERI-RB-1 cells and retinoblastoma tumors that migrate along the optic nerve in humans, but they do not show the migration route in the injected zebrafish either. Further, their analysis of cell migration ended 4 days after cell injection in the zebrafish eye. For this reason, our study is a good complement for further analysis of retinoblastoma in the zebrafish, because we showed that the migration of injected cells in the zebrafish eye is comparable with the metastasis of retinoblastoma cells in humans.

Concerning the results of Madreperla et al. (1991) [17], we could not reproduce their results regarding RB355 being a subclone of Y79. Our results on the sequencing of two alleles that were often mutated in retinoblastoma and on the MYCN status for 10 retinoblastoma cell lines were published in Schwermer et al., 2019 [13]. The MYCN amplification status was analyzed, because it was published that in a subgroup of retinoblastoma, no RB1 mutation was found, but instead a MYCN amplification. Therefore, we used in our study WERI-RB-1 cells, which have a detected RB1 deletion and no copy number of RB1. For a comparison of the results with this cell line, we used RB355 cells, which have a C > T mutation on exon 19 and a second T > A mutation at exon 22 and Y79 cells with a deletion in exon 2–6 and a G > A mutation at exon 20, both cell lines with a MYCN amplification of more than fourfold. We wanted to analyze if these cell lines show differences in the migration routes and which cell line shows the longest observable time by using GFP-marked cells. In our present study, we get much better results with RB355 cells than with Y79 cells because the RB355 cell line showed a longer observation time in our experiments. Therefore, further analysis (possibly for drug testing) with the zebrafish model of the RB355 cell line may show better results than Y79 cells, although both cell lines have an MYCN amplification status.

However, within the injections of one cell line we have observed a high variability of the fluorescence intensity on the days post injection from one fish to another, which limits the zebrafish model’s use in drug testing. Janostiak et al. described in a review in 2022 [28] the high complexity of the regulation of the function of the RB1 protein. This complex regulation of RB1 function includes E2F-dependent and E2F-independent signaling and also post-translational modifications of RB1 by phosphorylation, acetylation and methylation and many further regulatory factors. Despite the inter-experimental variability, we present for the first time a bone fide model of retinoblastoma in zebrafish that recapitulates the metastatic spread of the human disease. Thus, our model could serve as a starting point to better understand progression and metastasis of retinoblastoma.

## 5. Conclusions

In zebrafish, retinoblastoma cells injected into the eye migrated along the optic nerve and into the brain and brain ventricles. These routes of metastases were also found in retinoblastoma-affected children [6]. Thus, the zebrafish is suitable as an animal model for retinoblastoma research with short development time, small maintenance costs, clearness and transparency for microscopy analyses for a long time, and the metastasis route resembles that in human patients. However, drug testing for retinoblastoma medication may turn out to be difficult using the zebrafish model because of the restricted duration of a sufficiently high level of fluorescence intensity of the injected retinoblastoma cells. Alternatively, H&E staining of histological sections can be used to detect the trajectories of tumor cells in the host for a longer time.

## Figures and Tables

**Figure 1 cancers-14-05814-f001:**
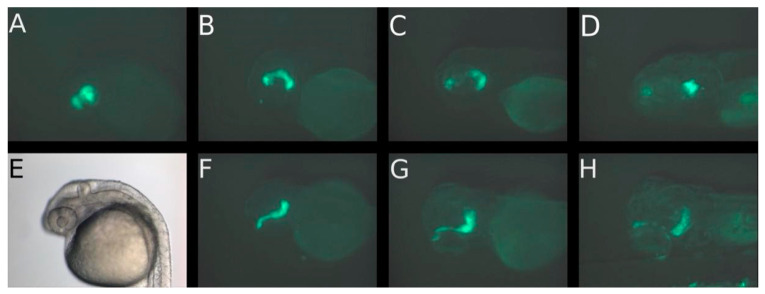
Migration of WERI-RB-1 cells in the first 5 days after injection of 80–100 suspension cells into the vitreous cavity of the left eye of a 2-day-old zebrafish (**A**) on day of injection, (**B**) on day 1, (**C**) on day 2 and (**D**) on day 5 post injection in lateral view; (**E**) on the day of injection with transmitted light, (**F**) on day 1, (**G**) on day 2 and (**H**) on day 5 post injection in dorsal view. Only the dorsal view shows the initiation of migration of WERI-RB-1 cells in the brain already after one day post injection. After 5 days post injection, only a few cells are found in the eye and most have migrated to the brain. On day 8 post injection no tumor cells were visible (not shown).

**Figure 2 cancers-14-05814-f002:**
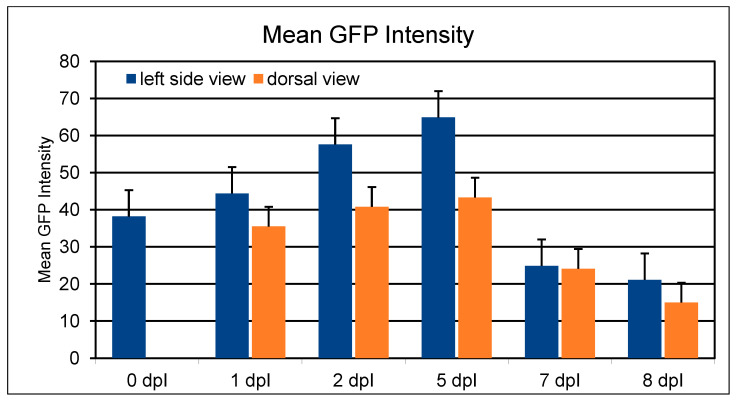
Mean GFP-fluorescence intensity of 20 zebrafish injected with RB355 cells in comparison of the left side view with the dorsal view. Determination of GFP-intensity was done with microscopic photographs from the left side view (left eye injected) and the dorsal view with the software ImageJ by marking the tumor margins. One zebrafish died at 1 dpi; 19 RB355-injected zebrafish could be observed until day 5 post injection (5 dpi) but the tumor cells could be observed until day 7 only in 10 cases. In cases where no fluorescence could be measured, the quantity was set to 12, which was the quantity of the background. The amount of analyzed zebrafish with measured fluorescence is given for each column in parentheses: 0 dpi (20), but could not be measured for dorsal view, 1 dpi (19), 2 dpi (19), 5 dpi (19), 7 dpi (10) and on 8 dpi (9), only one zebrafish could be measured on 9 dpi (not shown). Bars represent the standard error of measured fluorescence.

**Figure 3 cancers-14-05814-f003:**
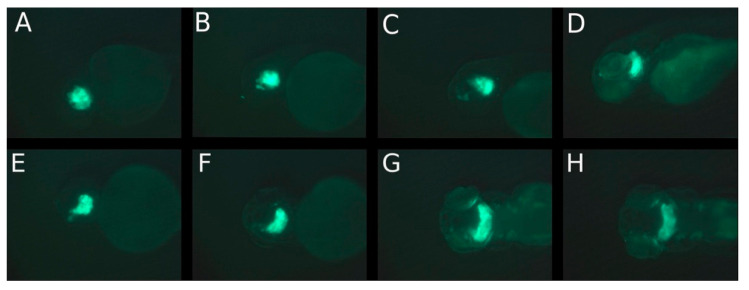
Positions of WERI-RB-1 cells in the first 5 days after injection of 80–100 suspension cells into the vitreous cavity of the left eye of a 2-day-old zebrafish (**A**) on day of injection, (**B**) on day 1, (**C**) on day 2 and (**D**) on day 5 post injection in lateral view; (**E**) on day 1, (**F**) on day 2, (**G**) on day 5 and (**H**) on day 8 post injection in dorsal view. The time interval includes the complete migration of the WERI-RB-1 cells out of the left eye into the brain.

**Figure 4 cancers-14-05814-f004:**
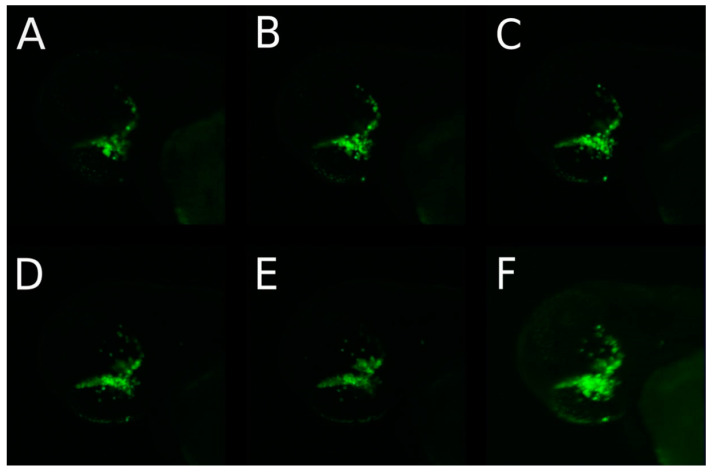
Retinoblastoma cells migrated towards the contralateral eye but not outside of the central nervous system. (**A**–**E**) shows 5 pictures of a z-stack in the dorsal head view taken with the Zeiss LSM 800 confocal microscope one day post injection (1 dpi) of 80–100 Y79 cells in the left eye. (**F**) shows the maximum intensity projection (MIP) of all five layers. The cells start to migrate from the retina of the left eye postulated along the optic nerve and reached near the right eye.

**Figure 5 cancers-14-05814-f005:**
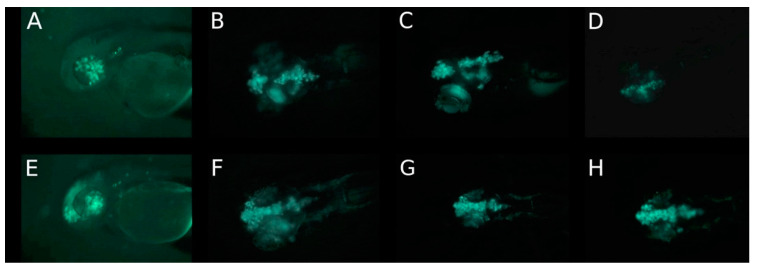
Retinoblastoma cells migrate into the ventricles at longer observation. Injection of 80–100 RB355 cells in a 2-day-old zebrafish eye. (**A**) on day of injection in lateral view, (**B**). on day 5, (**C**) on day 7 and (**D**) on day 9 post injection in dorsal view. (**E**) a second zebrafish on day of injection in lateral view, (**F**) on day 5, (**G**) on day 7 and (**H**) on day 9 post injection in dorsal view. Both injections show a migration of tumor cells into the ventricles after 5 days post injection. From 5 dpi until 9 dpi, the cells marked the ventricles of the forebrain, midbrain and hindbrain.

**Figure 6 cancers-14-05814-f006:**
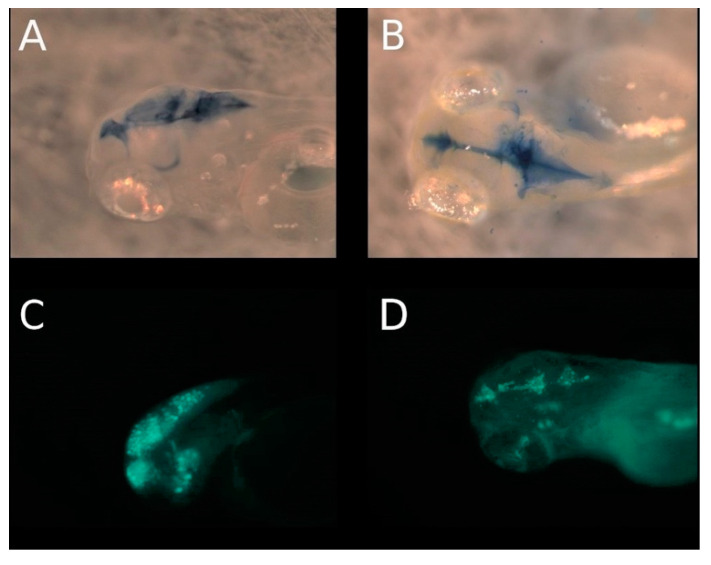
Demonstration of ventricles through direct cell-injection into the hindbrain ventricle. (**A**,**B**) demonstrates the injection of a methylene blue solution directly into the hindbrain ventricle. (**A**) shows a side view, (**B**) shows the dorsal view and (**C**,**D**) demonstrates the ventricles through direct injection of fluorescence-marked RB355 cells into the hindbrain ventricle of three-day-old zebrafish. The pattern of cell distribution after direct injection into the hindbrain ventricle (**C**,**D**) reflects the pattern of cell distribution after injection of the cells in the eye and postulated metastasis into the ventricles of fore-, mid- and hindbrain (as shown in Figure 5B–D,F–H and Figure 7B).

**Figure 7 cancers-14-05814-f007:**
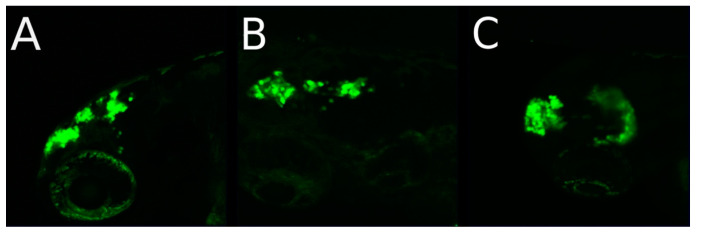
Metastatic spread of retinoblastoma cells into the brain ventricles. Injection of 80–100 Y79 cells into the left eye of a 2-day-old zebrafish. Figures are showing lateral and dorsal views by the confocal microscope Zeiss LSM800. Both views show metastasis of the tumor cells in the ventricle already after 4 days post injection: (**A**) on day 4 post injection in lateral view; (**B**) on day 5 post injection in dorsal view. The trapezoid shape of the forebrain ventricle after cell migration into this ventricle is visible (**A**,**B**) are from the same embryo). (**C**) a second embryo 4 days post injection showing one migration event in the anterior and a second one into the posterior region of the head. In both embryos, no cells can be seen in the injected eye after 4 days.

**Figure 8 cancers-14-05814-f008:**
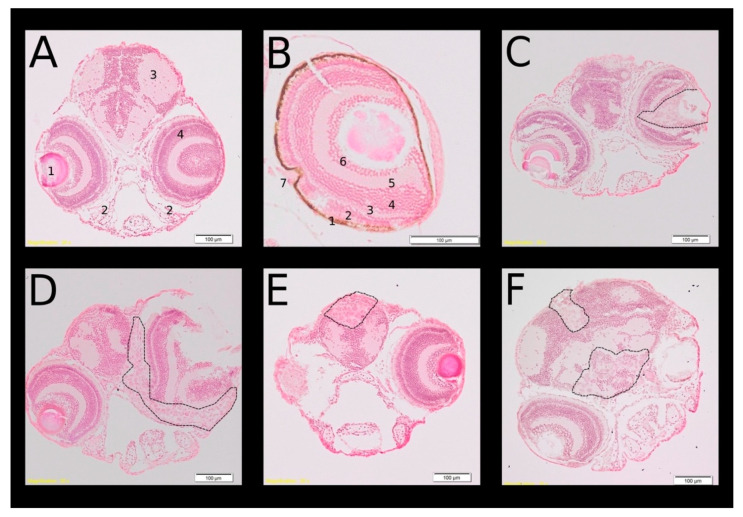
Hematoxylin-Eosin (H.E.) staining confirmed the localization of metastatic retinoblastoma cell line cells. Figure 8 shows frontal sections of different zebrafish heads in the region of the retina (**A**,**B**) Without cell injection and (**C**–**F**) after RB355 cell injection. (**A**,**C**–**F**) With objective magnification 20× and one eye; (**B**) in higher magnification (40×). (**A**) Zebrafish head of a six-day-old zebrafish without cell injection and the following marked structures: 1. lens, 2. pharyngeal cartilage, 3. brain at the forebrain-midbrain margin, 4. retina. (**B**) Retinal layers of a 6-day-old zebrafish: 1. retinal pigment epithelium (RPE), 2. photoreceptor layer (PRL) and outer nuclear layer (ONL), 3. outer plexiform layer (OPL) 4. inner nuclear layer (INL), 5. inner plexiform layer (IPL), 6. ganglion cell layer (GCL), 7. exit of the optic nerve (ON). (**C**) Two-day-old zebrafish at the day of RB355 cell injection, RB355 cells are visible only in the vitreous cavity. (**D**) Four-day-old zebrafish, two days after RB355 cell injection. Tumor cells have left the vitreous cavity and margin of the eye cup, and have migrated towards the brain. (**E**) Seven-day-old zebrafish, five days after RB355 cell injection. RB355 cells have entered the forebrain ventricle. (**F**) Nine-day-old zebrafish, seven days after RB355 cell injection. RB355 cells have entered the forebrain ventricle and some are visible near the location of the optic nerve and optic chiasma. (**C**–**F**) Injected RB355 cells are marked with a dashed line.

**Figure 9 cancers-14-05814-f009:**
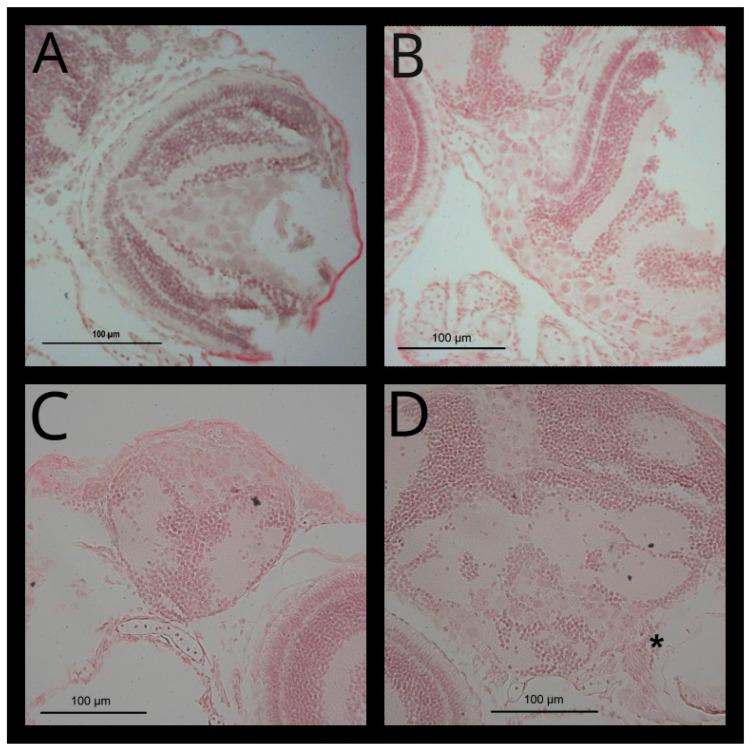
Higher magnifications of Figure 8C–F showing migration of injected RB355 cells in detail. (**A**) Two-day-old zebrafish at the day of RB355 cell injection. RB355 cells can be identified by the much larger size compared to cells of the retina layers of the host. Most of the cells are found in the vitreous cavity and some have migrated up to the photoreceptor layer. (**B**) Two days after RB355 cell injection, most of the injected cells have migrated through the retina of the injected eye and are situated at the margin of the epithelium of the eye cup. Some cells have migrated near the contralateral eye, but do not enter the contralateral eye (**C**) Five days after RB355 cell injection, migrated cells are found in the forebrain ventricle. (**D**) Seven days after RB355 cell injection. Asterisk marks axons of the optic nerve. RB cells are found immediately near the axons and migrated into the brain near to the contralateral eye, but have not entered the un-injected eye either. A second group of cells is visible in the forebrain ventricle.

## Data Availability

All data of reported results are saved on the personal computer from Nenad Maricic at the Department of Anatomy and Molecular Embryology, Ruhr-University Bochum, 44801 Bochum, Germany.

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
