# Peer review of "Zebrafish as an Orthotopic Tumor Model for Retinoblastoma Mimicking Routes of Human Metastasis"

_cancers, 2022, doi:10.3390/cancers14235814_

Round 1
Reviewer 1 Report
Maricic and colleagues describe the use of zebrafish to model retinoblastoma orthotopic xenografts, focusing in particular on how tumor spread in these animals mimics routes common in humans. In their zebrafish model, metastatic retinoblastoma cells were found “along the optic nerve and adjacent regions of the brain or its ventricles”, which mimics one aspect of spread seen in humans. The paper is well written, and, in general, clearly illustrated. However, it is almost entirely descriptive, with no mechanistic/molecular or therapeutic observations on retinoblastoma spread in fish.
Even the core observations on optic nerve and brain invasion only break limited new ground. Jo et al were the first to report on orthotopic retinoblastoma xenografts in fish (PMID: 23835085). Chen and colleagues subsequently published on the metastatic spread of retinoblastoma xenografts in zebrafish in 2015 (PMID: 26169357). They discussed spread along the optic nerve “Interestingly, approximately 90% of metastatic tumor cells were located in the head region of the zebrafish body, suggesting that cancer metastasis along the optic nerve and local invasion might be dominant metastatic pathways of retinoblastoma spreading.” The claim in the current submission that “Our study is the first one that describes the migration route of the retinoblastoma cells in the zebrafish” is therefore not entirely accurate, although Maricic et al do provide a more detailed description of this process.
A strength of the study is the histopathological confirmation retinoblastoma spread in tissue using microscopic analysis. However, higher magnification images are needed in figure 8, as in the current ones the presence of tumor cannot be confirmed or distinguished from normal structures. If necessary, immunohistochemical studies using antibodies specific for human epitopes (or GFP) could assist with this.
Another issue is the cell lines used. It has been suggested that RB355 is a subline of Y79, possibly generated via contamination (PMID: 1679230). The authors do not indicate in their methods section if identity testing has been performed, and it would seem critical for them to include STR testing on the lines they use to confirm that they are indeed independent of one another.
A final minor issue is omission of two references by Asnaghi and colleagues, whoe previously reported on spread of retinoblastoma Y79 orthotopic xenografts outside the eye (PMID: 31451106, PMID: 30401983). Although the path of spread including involvement of optic nerve and brain was not discussed in these, given the very limited literature in the area these prior references need to be discussed.
In summary, while understanding pathways of orthotopic retinoblastoma spread in zebrafish will be of some interest, and novel findings are included, overall the advance is rather descriptive and incremental, with no new mechanistic or molecular insights presented.
Author Response
Maricic and colleagues describe the use of zebrafish to model retinoblastoma orthotopic xenografts, focusing in particular on how tumor spread in these animals mimics routes common in humans. In their zebrafish model, metastatic retinoblastoma cells were found “along the optic nerve and adjacent regions of the brain or its ventricles”, which mimics one aspect of spread seen in humans.
The paper is well written, and, in general, clearly illustrated. However, it is almost entirely descriptive, with no mechanistic/molecular or therapeutic observations on retinoblastoma spread in fish.
Even the core observations on optic nerve and brain invasion only break limited new ground. Jo et al were the first to report on orthotopic retinoblastoma xenografts in fish (PMID: 23835085). Chen and colleagues subsequently published on the metastatic spread of retinoblastoma xenografts in zebrafish
in 2015 (PMID: 26169357). They discussed spread along the optic nerve “Interestingly, approximately 90% of metastatic tumor cells were located in the head region of the zebrafish body, suggesting that cancer metastasis along the optic nerve and local invasion might be dominant metastatic pathways of
retinoblastoma spreading.” The claim in the current submission that “Our study is the first one that describes the migration route of the retinoblastoma cells in the zebrafish” is therefore not entirely accurate, although Maricic et al do provide a more detailed description of this process.
Thank you for raising this point. We agree with the reviewer that in the first version of our manuscript we did not show directly the migratory route of tumor cells along the optic nerve in the H.E.-stained sections. While Chen et al. do not focus on the metastasis route of the injected tumor cells along the optic nerve at all, we now included a new figure from neighboring H&E-stained sections in higher magnification showing tumor cells adjacent to the axons of the optic nerve (Fig. 9D).
We have now changed the sentence in the revised version of our manuscript: “Our study is the first one that focuses in detail on the migration route of the retinoblastoma cells in the zebrafish along the optic nerve and into the ventricles.” (page 11).
A strength of the study is the histopathological confirmation retinoblastoma spread in tissue using microscopic analysis. However, higher magnification images are needed in figure 8, as in the current ones the presence of tumor cannot be confirmed or distinguished from normal structures. If necessary,
immunohistochemical studies using antibodies specific for human epitopes (or GFP) could assist with this.
Thank you for your comment. We now added higher magnifications of the Figures 8C-F in the manuscript as Figure 9A-D (page 11).
Another issue is the cell lines used. It has been suggested that RB355 is a subline of Y79, possibly generated via contamination (PMID: 1679230). The authors do not indicate in their methods section if identity testing has been performed, and it would seem critical for them to include STR testing on the lines they use to confirm that they are indeed independent of one another. Citation form Madreperla et al. (1991): “Interpretation of these and other published data suggest that both RB355 and WERI-Rb27 are probably sublines of Y79."
Thank you for your comment and the evidence on this publication. We have now added a part in the methods on our STR analysis (2.1. page 3) and a second part in the discussion on this (page 13).
A final minor issue is omission of two references by Asnaghi and colleagues, whoe previously reported on spread of retinoblastoma Y79 orthotopic xenografts outside the eye (PMID: 31451106, PMID:30401983). Although the path of spread including involvement of optic nerve and brain was not discussed in these, given the very limited literature in the area these prior references need to be discussed.
Thank you for your comment. We now included the two publications into the discussion (page 13).
In summary, while understanding pathways of orthotopic retinoblastoma spread in zebrafish will be of some interest, and novel findings are included, overall the advance is rather descriptive and incremental, with no new mechanistic or molecular insights presented.
Reviewer 2 Report
In this manuscript, Maricic et al. present a careful study of the progression of retinoblastoma xenografts in zebrafish. Three human cell lines are injected intraocularly and tracked over a period of up to eight days by fluorescence imaging, including some very nice intravital confocal imaging, plus histology. Fluorescence increased for several days before degrading rapidly. Migration of cells into the brain and ventricles, and possibly along the optic nerve, was documented. These findings expand on prior retinoblastoma zebrafish xenograft work through longer duration and both dorsal and lateral imaging, and the careful documentation and presentation provides useful information for those interested in using this model, including a caveat for future drug studies due to the variability of response. However, there are some major and minor issues that should be addressed:
Major concerns:
1. Since intraocular fluorescence decreases as extraocular site fluorescence increases, “metastases” may not be the correct term for what is more like a migration event. Language needs to be revised to acknowledge this important limitation.
2. Migration along the optic nerve is not conclusively shown; H&E showing this (or colocalization with an RGC axonal marker) is needed or else statements related to this in the Discussion need to be reworded.
Minor issues:
3. p. 2 “the time required for tumor development is significantly longer compared to human retinoblastoma.” This statement should be clarified to indicate that it is relative to the lifespan of the organism.
4. p.2, the authors should cite another pair of zebrafish xenograft papers with Y79 cells (PMID: 30401983 and doi: 10.1186/s40478-019-0785-4), although this was a short-term model
5. Section 2.2, please give more details on the electroporation conditions
6. Section 2.6, was fluorescence quantification volumetric (from Z-stacks), or from 2D images? If only from 2D images, it is important to acknowledge this limitation
7. p.5, “In all three cell lines from 20 injected zebrafish we have one zebrafish that died until the first day post injection.” Wording here is confusing; Table S1 indicates one death per line, i.e. 3 total.
8. Fig. 2, please indicate the n for each bar since it varies considerably. What do the error bars represent?
9. Fig. 4 caption mentions Fig. 3F, but Fig. 4F is I think intended
10. The discrepancy in findings between this study and Chen et al. 2015 regarding route of metastasis warrants some further discussion
11. p. 11, last discussion paragraph, suggestions about experimental variability due to pRB’s varied functions is purely speculative and should be removed
Author Response
Comments and Suggestions for Authors
In this manuscript, Maricic et al. present a careful study of the progression of retinoblastoma xenografts in zebrafish. Three human cell lines are injected intraocularly and tracked over a period of up to eight days by fluorescence imaging, including some very nice intravital confocal imaging, plus histology. Fluorescence increased for several days before degrading rapidly. Migration of cells into the brain and ventricles, and possibly along the optic nerve, was documented. These findings expand on prior retinoblastoma zebrafish xenograft work through longer duration and both dorsal and lateral imaging, and the careful documentation and presentation provides useful information for those interested in using this model, including a caveat for future drug studies due to the variability of response. However, there are some major and minor issues that should be addressed:
Major concerns:
1. Since intraocular fluorescence decreases as extraocular site fluorescence increases, “metastases” may not be the correct term for what is more like a migration event. Language needs to be revised to acknowledge this important limitation.
Thank you for your important comments. According this consideration in the corrected manuscript version we use the term “migration” instead of the term “metastases” for the injected cells.
2. Migration along the optic nerve is not conclusively shown; H&E showing this (or colocalization with an RGC axonal marker) is needed or else statements related to this in the Discussion need to be reworded.
Thank you very much for raising this issue. We have now included a new figure (Fig. 9A-D) with higher magnifications of slides from Figure 8C-F or adjacent slides and added the following aspect in the discussion. Figure 9D shows tumor cells adjacent to the optic nerve (marked with an asterisk) in an adjacent section demonstrating the migratory route more clearly. Because the optic nerve of a young zebrafish is very thin and does not run plane in one layer to the optic chiasma it is mostly not possible to get the whole optic nerve in one slide. Therefore, injected retinoblastoma cells can only be shown on a short distance in the slides. For this reason, we use both methods, the H.E.-stained slides and confocal microscopy on GFP-transfected cells with the Z-stack of five planes (pages 11 and 12).
Minor issues:
3. p. 2 “the time required for tumor development is significantly longer compared to human retinoblastoma.” This statement should be clarified to indicate that it is relative to the lifespan of the organism.
We corrected the sentence as following:“...the time required for tumor development is significantly longer compared to human retinoblastoma in comparison to the lifespan of the zebrafish.”
3. p.2, the authors should cite another pair of zebrafish xenograft papers with Y79 cells (PMID: 30401983 and doi: 10.1186/s40478-019-0785-4), although this was a short-term model.
Thank you for this important indication, we have now included both articles in the discussion (citation 26, 27 on page 13).
5. Section 2.2, please give more details on the electroporation conditions.
Thank you for showing this deficiency, we have now included a more detailed description on the transfection method for producing the GFP fluorescence cells (2.2. on page 3).
6. Section 2.6, was fluorescence quantification volumetric (from Z-stacks), or from 2D images? If only from 2D images, it is important to acknowledge this limitation.
Fluorescence quantification was made from the 2D images, because to do this with z-Stack at the confocal microscope with each zebrafish and for a time until eight days will be too time intensive and it will harm the young fish in addition (We added this point on 2.6. page 4).
7. p.5, “In all three cell lines from 20 injected zebrafish we have one zebrafish that died until the first day post injection.” Wording here is confusing; Table S1 indicates one death per line, i.e. 3 total.
This is right, one zebrafish per line, makes 3 in the sum, in detail it can be seen in the S1 table.
8. Fig. 2, please indicate the n for each bar since it varies considerably. What do the error bars represent?
Error bars represent the standard error calculated with excel of n zebrafishes. The amount of zebrafish is now given for each column in the legend of figure 2 (at 1dpi from 19 injections, because one fish died until 1dpi).
9. Fig. 4 caption mentions Fig. 3F, but Fig. 4F is I think intended
Thank you for this correction, I have corrected this Figure legend, you are right here.
10. The discrepancy in findings between this study and Chen et al. 2015 regarding route of metastasis warrants some further discussion
Thank you for highlighting this important point. We have now complemented the discussion for this point on page 12.
11. p. 11, last discussion paragraph, suggestions about experimental variability due to pRB’s varied functions is purely speculative and should be removed
The variability in the fluorescence of the injected RB355 cells is shown in Figure 2 by the standard error lines for each column and each analyzed day post injection. We have rephrased the last discussion paragraph, now showing the complexity of the regulation of RB1 independent of our results.
Round 2
Reviewer 1 Report
The revised manuscript is significantly improved, and my issues have been addressed.
Reviewer 2 Report
In this revision, Maricic et al. carefully address all the reviewer comments. Three points should be addressed when finalizing the manuscript: 1. New Figure 9 should be cited in the text; 2. Mention of “metatstases” in Table S1 should be changed to migration to match the main text. 3. The new paragraph about Madreperla et al. and the genotypes of the cell lines is repeated in Section 2.1 and the Discussion. I suggest deleting it from Section 2.1.